# Effect of Sinus Perforation with Flaplessly Placed Mini Dental Implants for Oral Rehabilitation: A 5-Year Clinical and Radiological Follow-Up

**DOI:** 10.3390/jcm11154637

**Published:** 2022-08-08

**Authors:** Luc Van Doorne, Geert Hommez, Ewald Bronkhorst, Gert Meijer, Hugo De Bruyn

**Affiliations:** 1Department of Plastic, Oral and Maxillo-Facial Surgery, University Hospital Ghent, Corneel Heymanslaan 10, 9000 Ghent, Belgium; 2Dentistry, Prosthetic Dentistry, Endodontics, Oral and Maxillo-Facial Surgery, Private Clinic “Het Tandplein”, Bilkske 68, 8000 Brugge, Belgium; 3Faculty of Medicine and Health Sciences, Ghent University, Heymanslaan 10, 9000 Ghent, Belgium; 4Radboud Institute for Health Sciences, Radboud University Medical Centre, 6525 GA Nijmegen, The Netherlands

**Keywords:** mini dental implants, sinus perforation, implant overdenture rehabilitation

## Abstract

Background: Flaplessly placed one-piece mini dental implants (MDI) are an option to support maxillary overdentures. Evenly distribution of the implants over the atrophic alveolar process implies a risk of accidental sinus perforation in the posterior area which could induce sinus-related pathology. Methods: Thirty-one patients received 5–6 maxillary MDIs. Schneiderian membrane swelling was assessed with CBCT at the deepest point of the sinus in the mid-sagittal plane prior to surgery (baseline), after 2 and 5 years. Additionally, subjective patient-reported rhinosinusitis complaints, the effect of smoking, and gender differences were investigated. Results: Mean thickness of the Schneiderian membrane was 2.87 mm at baseline, 3.15 mm at 2 years, and 4.30 mm at 5 years in 27 of 31 initially treated patients. MDI perforation was detected in 21/54 sinuses. At 2 years, perforation length does not affect membrane thickness whereas baseline swelling does. In smokers, each perforated mm induced 0.87 mm additional swelling. After 5 years, the effect of baseline swelling becomes smaller whereas perforation length became statistically significant, with 0.53 mm increase for every perforated mm. The effect of smoking lost its significance. No relations between gender, membrane thickness changes, or subjective clinical sinus complaints and MDI perforation were found. Conclusion: Accidental MDI sinus perforation induces Schneiderian membrane swelling but does not interfere with clinical sinusal outcome after 5 years.

## 1. Introduction

One-piece mini dental implants (MDIs) are a feasible option to support an overdenture in the severely atrophic maxillae [1]. In particular, when combined with minimally invasive flapless surgery, it leads to an increased patient benefit in terms of quality of life and satisfaction [2]. Extensive bone grafting is avoided, thus evading the burden of an extra operation. Furthermore, affordability improves as the surgical intervention is time-saving and less costly. As such, this treatment protocol is also apt for frail, anxious, and geriatric patients [3] and lowers the barrier for treatment due to affordability. Whilst the MDIs add to functional retention, additional reduction of the palatal plate of the denture improves taste [4] and comfort. Removal of the palatal support requires an adequate distribution of the implants over the alveolar process. Damghani et al. concluded that under in vitro testing conditions, four dental implants with an inter-implant distance of 16 mm or greater substantially reduce the stress that is generated by the denture base on the hard palate [5]. Benzing et al. [6] demonstrated that from a biomechanical perspective, a spread-out arrangement of six implants in the maxilla results in an even better load distribution. Biomechanical conditions are of importance and could interfere with morphological changes in bone around dental implants. Wolff [7] referred to a relationship between the internal bone architecture remodeling and the loading direction. Stanford and Brand [8] focused on mechanical strain as the main parameter for bone remodeling. Above a certain strain threshold, bone formation is induced, while extremely low strains can cause bone resorption. Hence, extremely high strains can lead to bone fracture before new bone is created. Strain distribution and magnitude of loading in the cortical and cancellous bone tissues are investigated by finite element analysis [9]. It was found that different implant designs, implant positioning [10], implant placement depth, abutment angulation, length of cantilever arm, and the use of different prosthetic framework materials with different elasticity modulus (Young’s Modulus) could interfere with bone to implant health [11,12]. However, implants in the posterior region may be jeopardized because of inadequate bone height due to sinus pneumatization or the degree of alveolar ridge resorption that may preclude the conventional implant placement. Various sinus floor elevation (SFE) procedures are developed to cope with these drawbacks, among which the transcrestal lateral window technique and the transalveolar sinus floor elevation technique are the most popular. The conventional transcrestal or the lateral window sinus floor elevation (LSFE) method introduced by Tatum (1974) requires surgical access through the lateral wall of the maxilla, followed by elevation of the sinus membrane and insertion of a bone graft under direct vision. The transalveolar sinus floor elevation (TSFE) described by Summers (1994) is a surgical technique used to vertically increase the bone volume in the atrophic maxilla by applying increasing diameters of osteotomes through the implant preparation site. A bone height of at least 6 mm is indicative for TSFE. In case of residual bone height less than 5 mm, LSFE is recommended [13,14,15]. Unfortunately, these additional surgical procedures increase costs and morbidity risk, which contradicts the wish to simplify the treatment protocol for the frail geriatric population in need of more convenient denture retention. With respect to SFE, most of the available literature reports on sinus complications after the aforementioned reconstructive procedures [16]. The maxillary sinus is a paired pyramid-shaped paranasal cavity lined with thin respiratory ciliated epithelium that serves in the transportation of fluid secretions toward the ostiomeatal unit (OMU). This lining is called the Schneiderian membrane [17]. The integrity of the membrane and the patency of the OMU seems to be important for the avoidance of potential complications [16,18]. In the vicinity to the sinus, residual bone height and smoking were factors affecting dental implant success [16]. However, preoperative sinusitis and Schneiderian membrane rupture were the main risk factors for postoperative sinusitis after reconstructive implant surgery. In a recent retrospective study by Park et al. [19], a significant association of implant-related odontogenic sinusitis was described in relation to implants extruding more than 4 mm into the maxillary sinus. Ragucci et al. [20] reported on survival and the complication rate of perforated implants into the sinus cavity without regenerative procedures or graft material. No statistically significant difference in the survival rate was observed between perforated or non-perforated implants. Only swelling of the Schneiderian membrane was observed, which correlated to the length of the implant penetration. Perforations less or more than 4 mm led to 5.3% and 29.3% radiographically increased sinus membrane thickness, respectively, however, this difference was not statistically significant.

Van Doorne et al. [21] introduced the concept of maxillary MDI overdenture retention and described an implant survival of 78.4% after 3 years. This survival is lower than with conventional dental implants and explained by compromised bone conditions, non-invasive but a free-handed flapless procedure and early loaded non-connected MDIs. Despite failures, patient satisfaction and the improvement of oral health-related quality of life, defined by means of the OHIP-14 questionnaires [2], were still high. Perforations on implant level of the alveolar crest, the nasal, and sinus floor were detected in 21.8% of the implants (37/170 MDI) [22]. However, no effect was seen for the long-term implant survival up to 2 years, which is similar to more conventional treatment procedures [20]. Although the available literature [16,20] is promising regarding implant survival and patient-centered outcome, one could question whether the minimal invasive flaplessly placed MDIs, which are evenly distributed over the maxilla (Figure 1), could induce severe sinus-related pathology in case of sinus perforation in the posterior maxillary region.

Chronic rhinosinusitis is, foremost, a clinical diagnosis based on the triad of head and facial pain syndromes, nasal airway obstruction, and rhinorrhea. This clinical diagnosis is additionally supported by endoscopic and imaging findings [23], whereby the thickness of the Schneiderian membrane seems to be an important marker of sinusal reaction in case of sinusitis, infections, and mucosal irritations [24]. 

The aim of this prospective case series was to scrutinize the radiographic changes of the Schneiderian membrane thickness in relation to implant perforation length measured on CBCT images taken prior to surgery after 2- and 5-years follow-up. An example of the chronological CBCT investigations of the case presented in Figure 1 is depicted in Figure 2.

The null hypothesis was that length of implant perforation has no effect on the Schneiderian membrane thickness and that perforation does not necessarily lead to clinical chronic rhinosinusitis over time.

Additionally, the effect of smoking, gender, and clinical patient-reported rhinosinusitis complaints by means of the Rhino Sinusitis-Severity Score questionnaire (R-SSS) [25] were assessed. 

## 2. Materials and Methods

### 2.1. Treatment Protocol

This multicenter prospective cohort study involved two treatment-center locations where surgery was performed by the same maxillo-facial surgeon (LVD). The study was approved by the Ethical Committees of the General Hospital AZ ZENO Knokke-Blankenberge and the Ghent University Hospital. (Belgium, registration number B670201422937). All included patients provided an informed consent before treatment. A detailed description of the treatment protocol has been already published [21]. In brief, only subjects aged 50 years or older and in need of improvement in maxillary denture retention were included. Patients were excluded when they suffered from uncontrolled systemic diseases, immune system dysfunctions, were previously treated with maxilla-facial and alveolar bone reconstruction, or when they had a history of therapy involving oral or intravenous bisphosphonates or radiotherapy in the maxillofacial area. Smoking was not considered as a contraindication for treatment. Only patients without sinus complaints were included. The MDIs (ILZ, Southern Impl. Inc., Irene, South Africa) were one-piece tapered and made of pure titanium Class 4 with a moderately rough surface (Sa 1.5 μm) and a maximum coronal diameter of 2.4 mm and a threaded part of 10 or 11.5 mm long. The transmucosal part is 4.8 mm long, with a smooth surface (Sa 0.4 μm), and with a coronal ball of 1.8 mm diameter on top for prosthetic retention. 

Placement of six MDIs was performed under local anesthesia with a free-handed flapless approach and patients were advised to refrain from denture wearing until one week postoperatively. Hereafter, the denture was adapted with a retentive soft reliner. Patients received oral hygiene instructions and were followed on a regular basis by either the surgeon or prosthodontist. The final prosthetic connection, with a new metal-reinforced horse-shoe denture and activated attachment clips onto the ball parts of the MDI, was established at least 6 months following implant placement. 

### 2.2. Radiological Examination

The CBCT was taken with Planmeca Promax™ 3D dental CBCT and analyzed with Planmeca Romexis™ software (Planmeca Oy, Helsinki, Finland). Examination was performed with the patient in a stabilized sitting position. First, a scout image was performed to scrutinize acceptable horizontal maxillary jaw positioning before final DICOM image execution. The specific imaging protocol used for the maxilla comprised a voxel size of 200 μm, a field of view of 80 mm × 50 mm, 90 KV, and a variable range of 5–8 mA depending on patient morphology/gender. The exposure time was 12 s, thereby reducing the patient’s dose to a minimum, according to the “as low as reasonably achievable” (ALARA) principle [26]. 

Janner et al. [27] described the dimensions of the Schneiderian membrane as depicted by CBCT in patients referred for dental implant surgery in the posterior maxilla. The highest statistically significant values of swelling were found in the mid-sagittal plane cutting through the deepest point of the sinus floor in the coronal CBCT slice. These findings were explained by the influence of gravitational forces in a seated patient position during CBCT image taking. This CBCT examination method for calibrated measurements is demonstrated in Figure 3.

The thickness of the Schneiderian membrane was chronologically assessed, prior to implant placement (baseline), post-operatively after 2 years, and 5 years.

Additionally, the following measurements were registered for each sinus: presence of a perforation and perforation length. All measurements were performed twice by one experienced examiner (L.V.D). When the difference between two values was larger than 0.2 mm, a third measurement was performed as proposed by Bornstein et al. [28]. For calibration and evaluation of intra-observer reliability, in 10 randomly selected cases, measurements were repeated at two different timepoints, respecting a minimum time interval of 2 months.

### 2.3. Clinical Rhinosinusitis Evaluation

For assessing patients‘ subjective sinus condition, we used the symptom-driven Quality of Life questionnaire, which was developed by Malan JF and published by Rustogi et al. [25], defined as the Rhinosinusitis Symptom Severity Score (RS-SSS). This methodology assesses the overall severity of chronic rhinosinusitis (CRS) at one moment in time in conjunction with a radiological Rhinosinusitis typing (RST) system. It is based on 15 parameters limited to six major symptoms being: (1) nasal congestion/blockage, (2) clear nasal discharge, (3) discolored nasal discharge, (4) facial pain, (5) sinus headache, (6) upper jaw toothache and nine additional minor symptoms. The latter are: (7) disturbed smelling, (8) cave-like speech, (9) sore throat, (10) fever above 37.8 °C, (11) foul breath, (12) swelling around eyes, (13) cough, (14) wheezy chest, and (15) tiredness. Each symptom has to be scored by the patient on a Likert scale whereby 0 = symptom free; 1 = some discomfort; 2 = moderate discomfort, which sometimes interferes with daily activities and sleep; 3 = severe discomfort, regularly preventing participation in work, school, and other activities. To qualify, a patient must have at least one major symptom. The overall outcome of RS-SSS is composed by summation of the score of symptom 1; the highest score of symptoms 2 or 3; the highest score of symptoms 4, 5, or 6; and finally, the highest score of symptoms 7 to 15 wherein the latter must be divided by 3. The obtained value is distributed in different categories with 1 to 2.66 scored as mild; 4 to 8.66 scored as moderate and 9 or more scored as severe. The RS-SSS in our study was assessed at 2 and 5 years postoperatively. 

### 2.4. Statistical Analysis 

Statistical analysis was performed using SPSS version 25 (IBM SPSS, statistics for Windows, version 25.0, Business Analytics, Amonk, NY, USA) and R version 3.6.3. The intra-observer performance was analyzed using a paired *t*-test, to check for systematic difference. The duplicate measurement error was calculated as the standard deviation of the differences divided by √2. Additionally, the reliability was calculated by the Pearson correlation coefficient. Change of the Schneiderian membrane thickness over time was analyzed using linear mixed models. To reduce the impact of individual outliers regarding membrane thickness, the highest values were winsorized at the 90th percentile. 

## 3. Results

In the initial study [21], 31 patients with dentate mandibles, and a mean age of 62.30 years (SD 9.28), received six maxillary MDIs for overdenture treatment. After 5 years, 27/31 (87%) of the patients (12/27 (44.4%) females; 15/27 (55.5%) males) with a total of 54 maxillary sinus sites, were available for assessment. Duplo analyses showed a reliability of 0.994. An absence of a systematic difference between measurements was observed as the duplicate measurement error was 0.814 mm; mean difference was 0.263 mm; 95% ciP [−0.276…0.801]; *p* = 0.321.

Mean thickness of the Schneiderian membrane prior to surgery (baseline) for all membranes was 2.87 mm (SD 4.2; range 0–20.6 mm). In 22/54 sinuses (40.7%), no swelling was detectable; in 11/54 (20.4%), swelling ranged between 0–2 mm; and in 21/54 (38.9%), above 2 mm. Baseline swelling above 10 mm was found in 3/54 sinuses (5.6%) with a maximum thickness of 20.60 mm in one sinus. 

At 2 and 5 years after implantation for all investigated membranes, the mean swelling was 3.15 mm (SD 4.46; range 0–18.4 mm) and 4.30 mm (SD 4.30; range 0–35.6 mm), respectively. In 21/54 (38.9%) sinuses, MDI perforations were detected, and 33/54 (61.1%) sinuses revealed no MDI perforations. In Table 1, the evolution of the mean thickness of the Schneiderian membrane is demonstrated for the whole group, and for the groups with and without perforation.

The effect of perforation length on membrane thickness at either 2 or 5 years was analyzed, using the membrane thickness at baseline as an independent variable in the regression model in addition to the length of the perforation in mm. Membrane thickness at baseline was centered around its mean value of 2.87 mm. Table 2 demonstrates that over a time interval of 2 to 5 years, the effect of the perforation length increases, while the size of the effect of thickness at baseline becomes smaller.

We analyzed whether smoking and gender were related to change in membrane thickness, again taking baseline membrane thickness (centered) and perforation length into account. Only the analyses for smoking are presented in Table 3, as no relation between gender and membrane thickness were found. In order to allow for a relation between the effect of smoking and the length of the perforation, an interaction term for these two factors was included in the model.

The results at 2 and 5 years do differ: at 2 years, in general, the perforation does not affect membrane thickness. Only in the smoking group, each mm of perforation is associated with an increase of 0.867 mm of membrane thickness, an interaction effect which diminishes to 0.351 mm at 5 years and thereby becomes statistically insignificant. In addition, smoking in general, irrespective of the presence of a perforation, was associated with an increase of membrane thickness by 1.526 mm at 2 years, which changed into a non-significant reduction of thickness by 1.140 mm at 5 years. At 5 years, the effect of the perforation is clearly significant, with an increase in thickness of 0.534 for every mm of perforation, while the interaction term and the effect of smoking in general have lost their statistical significance.

The final clinical outcome, as scored by the RS-SSS, related to the presence of perforation, gave a *p*-value of 1.000 at 2 years and 0.742 at 5 years by means of the Fisher Exact test. After 2 and after 5 years, the slight increase (Table 4) is remarkable, however not statistically significant. To conclude; no clear correlation is found between subjective RS-SSS complaints and MDI sinusal perforation.

## 4. Discussion

In an earlier paper [22], we reported a mean thickness of 0.64 mm (2.02 mm SD; 19.8 mm range) for the Schneiderian membrane after 2 years follow-up, in the vicinity of the apex of the MDIs. The present CBCT study prospectively investigated the evolution of thickness of the Schneiderian membrane, at an elaborately proposed location as described by Janner et al. [27]; namely, the mid-sagittal plane, which cuts through the deepest point of the sinus floor in the coronal CBCT slice, 2 and 5 years following placement of maxillary MDI in vicinity of the sinus. 

In 27 subjects with edentulous maxillae, as such representing 54 maxillary sinuses, the mean value of Schneiderian membrane swelling was 2.87 mm preoperatively, showing a large variability. 

In 38.9% of the sinuses, this swelling was above 2 mm. This is a much higher prevalence than the reported 17.73% (95% CI, 8.67–29.08%) in the systematic review by Razi et al. [29], scrutinizing 16.966 sino-nasal asymptomatic adults and children. They emphasized that this outcome was affected by confounding factors such as associated periodontal disease in dentate subjects, different type of imaging protocols, seasonal influence, the lack of standardization of the measurement method, and the measurement location. In contrast, our study measurements were standardized, as such allowing for assessments over time. 

The preoperative condition of all membranes, independent of the presence of a perforation, yielded an additional swelling after 2 and 5 years, with 49.8% increase after 5 years. However, when sinus perforation was present, swelling doubled and reached more than 134.97% of additional swelling after 5 years. In the systematic review of Ragucci et al. [20], the penetration of a normal size dental implant in the sinus cavity without augmentation also showed thickening of the sinus membrane which was interpreted as the most common radiographic complication without any clinical relevance, and without consequences for implant survival. Implant perforation length becomes an important factor for induction of additional membrane swelling over time, as corroborated by Jung et al. [30,31]. They stated that a minimal sinus membrane perforation of less than 2 mm heals spontaneously due to coverage of the implants with sinus mucosa or sometimes even bone [32], whereas implants penetrating the mucosa of the sinus floor by more than 4 mm can hardly be protected by the newly formed sinus membrane. Consequently, such implants are prone to accumulation of debris on the exposed surfaces, giving rise to possible sinus complaints. Raghoebar et al. [18] reported on a case of chronic rhinosinusitis, induced by two implants perforating the nasal base. Resection of the apical parts altered airflow, thereby reducing the complaints. A 2-year prospective observational study [33] of 19 patients treated for implant induced sinusitis, who underwent transcrestal SFE, revealed that it was not due to the implant itself, but the absence of patency of the ostiomeatal unit (OMU), and disturbance of the normal mucociliary function, that were the main reasons for sinusal pathology. Cause for these blockades were foreign bodies, bone graft materials, or granulation tissue. Troeltzsch et al. [34] reported on clinical characteristics of symptomatic unilateral maxillary sinusitis in 174 cases, 13 of which were associated with pre-implant surgery of dental implants. They reported that in all implant cases, maxillary sinusitis occurred as a consequence of advanced peri-implantitis after 4 years. Park et al. [10] evaluated 221 patients in a case-control study with 379 transcrestally placed implants, with a mean follow-up of nearly 10 years. They concluded that there were no significant differences in the thickness of the sinus membrane between the implants with/without perforation. Furthermore, no specific factors were identified to influence sinus membrane thickness, including perforation length more or less than 1 mm, which is refuted by our study.

Smoking is a bad habit inducing a variety of clinical pathologies, including impairment of mucociliary clearance of the respiratory system [35]. In the systematic review of Monje et al. [36], mean sinus membrane thickness for non-smokers did not exceed 1.05 mm, whereas smokers showed a mean value of 2.64 mm. They concluded that smokers possess thicker Schneiderian membranes compared with non-smokers. In our study, smoking, irrespective of implant perforation, induced an increase of Schneiderian membrane thickness of 1.526 mm after 2 years, with an unexpected decrease or atrophy of −1.140 mm after 5 years. Every mm of perforation was associated with an increase of membrane thickness of 0.867 mm after 2 years. Again, this effect seems to be reduced after 5 years with only a 0.351 mm increase of membrane thickness for every mm of perforation. 

Although the sinus membrane responded to the perforation, from a radiographical point of view, clinical symptoms for sinus pathology were lacking and subjective complaints were absent as confirmed by the RS-SSS score questionnaire. Despite several cases reporting severe membrane thickness, no statistic significant correlation between swelling and subjective reported clinical symptomatology over a 5-year period was observed. Whether the increase of membrane thickness between 2 and 5 years will further increase and lead to future complaints, remains to be investigated through longer follow-up. 

Our clinical study has several limitations which should be recognized. The main drawback is that the initially lost implants were not radiologically evaluated because they were removed prior to taking a CBCT. One cannot exclude that mispositioning or sinus mucosa perforations are associated with the encountered failures. 

## 5. Conclusions

The hypothesis has to be rejected that length of implant perforation has no effect on the Schneiderian membrane thickness over time. Accidental perforation induced Schneiderian membrane swelling, which was dependent on baseline swelling and the perforation length. However, it did not necessarily lead to chronic rhinosinusitis, nor interfere with the clinical sinusal outcome after 5 years follow-up. 

## Figures and Tables

**Figure 1 jcm-11-04637-f001:**
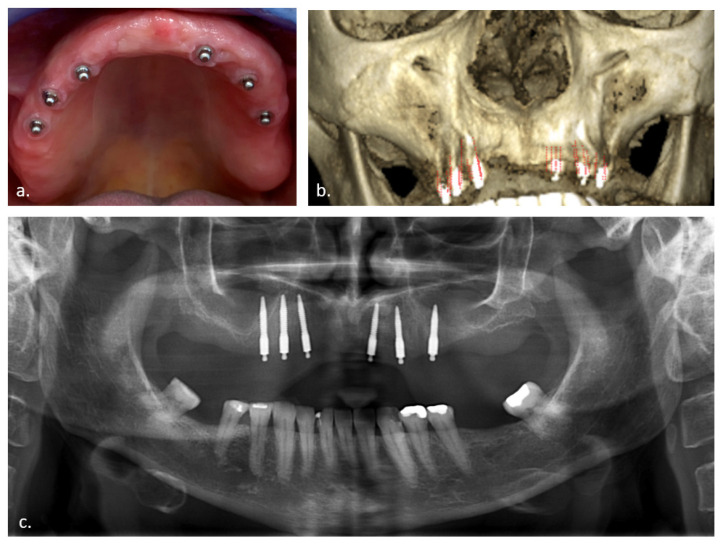
Case presentation: (**a**) Optimal distribution of one-piece mini dental implants (MDI) in the edentulous maxilla; (**b**) CBCT 3D presentation; (**c**) Panoramic 2D X-ray with posterior MDI perforation of the maxillary sinus.

**Figure 2 jcm-11-04637-f002:**
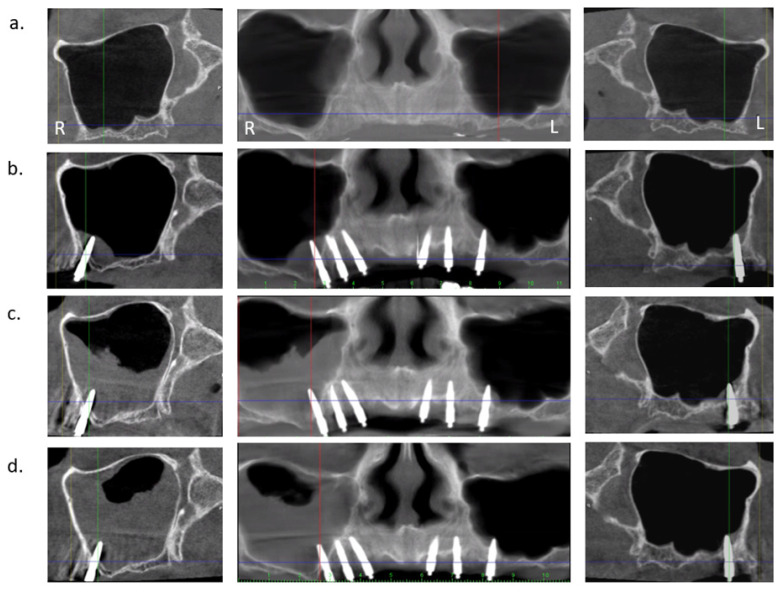
Chronological CBCT assessment of patient in Figure 1: MDI sinusal perforation on both sides with progressive thickening of the Schneiderian membrane at the right side (R) and absence of swelling at the left side (L); (**a**) before surgery (baseline), (**b**) 2 years after surgery, (**c**) 3 years after surgery, and (**d**) 5 years after surgery.

**Figure 3 jcm-11-04637-f003:**
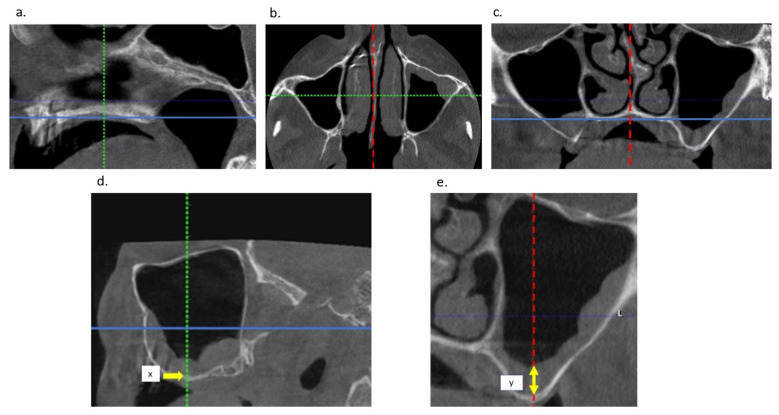
CBCT positioning for calibrated measurements of Schneiderian membrane thickness (Janner et al. [27]): (**a**) Sagittal view: nasal base is positioned parallel with the horizontal plane; (**b**) Axial view: the green line is the cutting slice (x) where we measured membrane thickness at the deepest basal point of the sinus (y), with the nasal septum perpendicular positioned on the horizontal plane; (**c**) Coronal view: nasal septum perpendicular on nasal base; (**d**) Demonstrating measurement position x in the sagittal plane; (**e**) Demonstrating measurement y in the coronal plane.

**Table 1 jcm-11-04637-t001:** Mean thickness (mm) of Schneiderian membrane after 2 and 5 years is presented either with or without MDI perforation.

Membrane Thickness (mm)	All Membranes (*n* = 54)	No Perforation (*n* = 21)	with Perforation (*n* = 33)
**Baseline**	2.87 (SD: 4.20; range 0.0–20.6)	2.49 (SD: 4.46; range 0.0–20.6)	3.46 (SD: 3.78; range 0.0–11.2)
**After 2 years**	3.15 (SD: 4.46; range 0.0–18.4)	2.31 (SD: 4.29; range 0.0–18.4)	4.49 (SD: 4.51; range 0.0–15.4)
**After 5 years**	4.30 (SD: 7.13; range 0.0–35.6)	1.87 (SD: 2.92; range 0.0–14.6)	8.13 (SD: 9.81; range 0.0–35.6)

**Table 2 jcm-11-04637-t002:** Effect of perforation length and baseline swelling on membrane thickness after 2 and 5 years.

2 Year	Estimate	95% Confidence Interval	*p*-Value
Intercept	2.607	[1.905…3.325]	<0.001
Thickness baseline centered (mm)	0.782	[0.612…0.955]	<0.001
Perforation length (mm)	0.212	[0.003…0.425]	0.051
**5 Years**			
Intercept	2.163	[1.284…3.038]	<0.001
Thickness baseline centered (mm)	0.597	[0.381…0.812]	<0.001
Perforation length (mm)	0.638	[0.378…0.897]	<0.001

**Table 3 jcm-11-04637-t003:** Separate effect of baseline swelling (centered)/perforation length/smoking and combined effect of perforation length and smoking on membrane thickness after 2 and 5 years.

2 Year	Estimate	95% Confidence Interval	*p*-Value
Intercept	3.122	[2.444…3.800]	<0.001
Thickness baseline centered (mm)	0.827	[0.678…0.986]	<0.001
Perforation length (mm)	0.000	[−0.201…0.203]	0.999
Smoking (yes vs. no)	1.526	[−2.673…-0.379]	0.019
Perforation length and smoking	0.867	[0.480…1.274]	<0.001
**5 Year**			
Intercept	2.556	[1.511…3.601]	<0.001
Thickness baseline centered (mm)	0.605	[0.394…0.818]	<0.001
Perforation length (mm)	0.534	[0.233…0.830]	0.001
Smoking (yes vs. no)	−1.140	[−2.917…0.636]	0.236
Perforation length and smoking	0.351	[0.378…0.897]	0.251

**Table 4 jcm-11-04637-t004:** Rhinosinusitis Symptom Severity Score (RS-SSS) questionnaire, 2- and 5-year outcome with or without perforation.

Complaints 2 Years	No Perforation (*n*: 10)	Perforation (*n*: 13)	Total (*n*: 23)
no	7 (70.0%)	8 (61.5%)	15 (65%)
mild	2 (20.0%)	3 (23.1%)	5 (21.7%)
moderate	1 (10.0%)	2 (15.4%)	3 (13.0%
severe	0 (0.0%)	0 (0.0%)	0 (0.0%)
**Complaints 5 Years**			
no	6 (54.5%)	6 (45.2%)	12 (50.0%)
mild	4 (36.4%)	4 (30.8%)	8 (33.3%)
moderate	1 (9.1%)	3 (23.1%)	4 (16.7%)
severe	0 (0.0%)	0 (0.0%)	0 (0.0%)

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
