# Peer review of "Effect of Sinus Perforation with Flaplessly Placed Mini Dental Implants for Oral Rehabilitation: A 5-Year Clinical and Radiological Follow-Up"

_jcm, 2022, doi:10.3390/jcm11154637_

Round 1

Reviewer 1 Report

The following manuscript has been submitted for a review:"Effect of sinus perforation with flaplessly placed mini dental implants for oral rehabilitation: A 5-year clinical and radiological follow-up." English typing style is correct. Abstract looks ok. Introduction could be shorten in some points, lines 90-104, while some improvements are necessary when the dental implant failure risks, lines 77-78, are lightly considered; they must be introduced and are depending on the bone mechanical response, for example, and the concept is not introduced. A more general introduction of this point is requested. Please fill the gap referencing it: (A) The role of cortical zone level and prosthetic platform angle in dental implant mechanical response: A 3D finite element analysis. Ausiello et coll. 2021. Dent Mater 37(11): 1688-1697; B) Influence of Framework Material and Posterior Implant Angulation in Full-Arch All-on-4 Implant-Supported Prosthesis Stress Concentration. Tribst J. et al 2022. Dentistry Journal.) Materials and Methods. They are well described. The CBCT analysis and methodology is deeply discussed and in line with its importance with the aim of the manuscript: "The aim of this prospective case series was to scrutinize the radiographic changes of the Schneiderian membrane thickness in relation to implant perforation length measured on CBCT images taken prior to surgery after 2- and 5- years follow-up." Results are well plotted. Statistics are ok. Discussion. It looks interesting. Results are scientifically discussed  and related each other, underlining also the limit this study anyway still have. In the conclusions the Authors rejected the hypothesis. Could you gently focus also in the Introduction this concept? Bibliography lloks up-graded. Replace the reference n. 1 with another Author on the same topic, please.

Reviewer 2 Report

The authors presented an interesting topic and a well-written manuscript. The study protocol appears to be well thought-out and sound.

In general, the position of the citation-numbers should be revised as it is partially at the end of the phrase and partially directly behind the mentioned author, e. g. LL 63-67.

Abstract:

L 24: I recommend that the authors put the abbreviation for mini dental implants directly to the place where they first mention it in order to offer a better readability.

Material and Methods:

It would have been interesting if the RS-SSS would have been assessed initially before the surgeries in order out whether any of the subjects enrolled for the study had any symptoms. When only symptom-free patients (considering signs of sinusitis) were enrolled for the study, maybe the authors could state this fact in the M&M section.

Figures:

Good quality and improving the understanding for the reader.

Tables:

Understandable without problems.

Author Response

Pleas see the attachment
